# CT-Angiography-Based Outcome Prediction on Diabetic Foot Ulcer Patients: A Statistical Learning Approach

**DOI:** 10.3390/diagnostics12051076

**Published:** 2022-04-25

**Authors:** Di Zhang, Wei Dong, Haonan Guan, Aobuliaximu Yakupu, Hanqi Wang, Liuping Chen, Shuliang Lu, Jiajun Tang

**Affiliations:** 1Department of Burn, Ruijin Hospital, Shanghai Jiao Tong University School of Medicine, Shanghai 200025, China; zd_1314521@sjtu.edu.cn (D.Z.); ghnsurg@163.com (H.G.); haximyakup@163.com (A.Y.); 2Wound Healing Department, Shanghai YongCi Rehabilitation Hospital, Shanghai 200025, China; wei.dong@aliyun.com; 3Department of Radiology, Ruijin Hospital, Shanghai Jiao Tong University School of Medicine, Shanghai 200025, China; whq01c35@rjh.com.cn (H.W.); clp01c73@rjh.com.cn (L.C.)

**Keywords:** diabetic foot ulcer, artificial neural networks, lower extremity CT angiography

## Abstract

The purpose of our study is to predict the occurrence and prognosis of diabetic foot ulcers (DFUs) by clinical and lower extremity computed tomography angiography (CTA) data of patients using the artificial neural networks (ANN) model. DFU is a common complication of diabetes that severely affects the quality of life of patients, leading to amputation and even death. There are a lack of valid predictive techniques for the prognosis of DFU. In clinical practice, the use of scales alone has a large subjective component, leading to significant bias and heterogeneity. Currently, there is a lack of evidence-based support for patients to develop clinical strategies before reaching end-stage outcomes. The present study provides a novel technical tool for predicting the prognosis of DFU. After screening the data, 203 patients with diabetic foot ulcers (DFUs) were analyzed and divided into two subgroups based on their Wagner Score (138 patients in the low Wagner Score group and 65 patients in the high Wagner Score group). Based on clinical and lower extremity CTA data, 10 predictive factors were selected for inclusion in the model. The total dataset was randomly divided into the training sample, testing sample and holdout sample in ratio of 3:1:1. After the training sample and testing sample developing the ANN model, the holdout sample was utilized to assess the accuracy of the model. ANN model analysis shows that the sensitivity, specificity, positive predictive value (PPV), negative predictive value (NPV) and area under the curve (AUC) of the overall ANN model were 92.3%, 93.5%, 87.0%, 94.2% and 0.955, respectively. We observed that the proposed model performed superbly on the prediction of DFU with a 91.6% accuracy. Evaluated with the holdout sample, the model accuracy, sensitivity, specificity, PPV and NPV were 88.9%, 90.0%, 88.5%, 75.0% and 95.8%, respectively. By contrast, the logistic regression model was inferior to the ANN model. The ANN model can accurately and reliably predict the occurrence and prognosis of a DFU according to clinical and lower extremity CTA data. We provided clinicians with a novel technical tool to develop clinical strategies before end-stage outcomes.

## 1. Introduction

Diabetic foot ulcer (DFU) is a common and serious complication of diabetes [1,2], which imposes a huge burden on patients and society [3,4,5]. DFU is the leading cause of lower extremity amputation (LEA) [6,7]. The development of a DFU is the result of a combination of multiple risk factors, of which the lower extremity vasculature is the most important aspect.

Prevention of DFU occurrence and recurrence is currently the focus of research. Independent risk factors associated with DFUs identified in current studies include: the vibration perception threshold >25 V, presence of a pre-ulcerative lesion, presence of peripheral artery disease, presence of an ulcer on the plantar foot, presence of a previous ulcer at the plantar hallux, presence of osteomyelitis, a Geriatric Depression Scale score ≥10, C-reactive protein >15 mg/L, glycated hemoglobin >7.5, loss of protective sensation, no in-shoe peak pressure <200 kPa and footwear adherence >80%, barefoot dynamic peak plantar pressure (per 100 kPa), day-to-day variation in step activity (per 100 strides), and cumulative duration of previous foot ulcers (per month) [1]. In previous clinical practice, the standard practices for DFU management included surgical debridement, dressings to promote a moist wound environment and exudate control, wound off-loading, vascular assessment and treatment, treatment of infection, and glycemic control [8]. In addition to the standard practices, a number of adjuvant therapies have emerged in recent years as the research process has evolved, including non-surgical debridement (such as autolytic debridement with hydrogels), dressings and topical agents (such as topical antiseptics and antimicrobials), oxygen therapies, acellular bioproducts, human growth factors, bioengineered skin, and energy-based therapies (such as electrical stimulation and laser therapy) [8,9].

Previous studies have found some biomarkers, such as procalcitonin, pentraxin-3, C-reactive protein (CRP) [10], erythrocyte sedimentation rate (ESR) [11], superficial hemoglobin (HbT1) and subsurface hemoglobin (HbT2) [12], to predict the onset of a DFU and amputation, which are often invasive or costly. Some clinical tests are commonly used in previous studies to predict the wound healing of diabetic foot ulcers and the risk of amputation, primarily transcutaneous oxygen measurement (TcPO2) and ankle–brachial index (ABI) [13]. Although they have the advantage of being non-invasive and radiation-free, they do have low predictive accuracy and require specialized equipment for measurement, which also increases the workload of clinicians [13]. In this study, 10 variables derived from clinical data and lower extremity computed tomography angiography (CTA), which were easily accessible in clinical practice, were selected as predictors. 

CTA is an increasingly attractive imaging modality due to rapid technological developments for assessing lower extremity arterial stenosis in current clinical practice [14,15]. Shorter acquisition times, thinner slices, higher spatial resolution, and improvements in multidetector computed tomographic scanners have made it possible to scan the entire vascular tree with fewer contrast materials in a limited period. Because of the noninvasive nature of the procedure, lower expenses (compare with digital subtraction angiography), and high accuracy (overall sensitivity and specificity rates of around 98%) [14,16,17], CTA is widely used in the assessment of hemodynamically significant stenosis and occlusion. 

In previous studies, the logistic regression (LR) is the most commonly used prediction model to identify the risk factors for and predict the occurrence of a DFU and LEA [18,19,20]. As a generalized linear model (GLM), LR is fast in model training and has good interpretability, but it struggles with a large number of multi-categorization features or variables, and it is prone to under-fitting, which affects the model accuracy. Recently, the artificial neural network (ANN) is widely used in different fields such as image processing, clinical diagnosis, and prognosis prediction [21]. Because of its ability to identify nonlinear interactions in a high-dimensional dataset, ANN models are gaining attention in the field of predictive modeling. The artificial neural network is a type of artificial intelligence that mimics the biological nervous system to process complex nonlinear data. The artificial neural network is made up of the input layer, the hidden layer, and the output layer (Figure 1). The input layer is composed of predictive factors derived from clinical data as neurons. The neurons in the hidden layer receive information from the input layer and are connected to the output layer, and the neurons between the layers are multiply connected by weights. The output layer forms the output.

Currently, there is no adequate study predicting the occurrence and prognosis of DFUs with an accurate model based on clinical data and lower extremity CTA. In the present study, 10 predictive factors (age, gender, body mass index, duration of diabetes mellitus, duration of a diabetic foot ulcer, limb symptoms, degree of lower extremity arterial stenosis, segment of lower extremity arterial stenosis, arterial calcification, and comorbidities) were derived from patients with diabetic foot ulcers based on the clinical data and lower extremity CTA data. The purpose of this study is to achieve accurate prediction of occurrence and prognosis of diabetic foot ulcers based on data easily obtained in clinical practice, using an artificial neuron network model. 

## 2. Methods and Materials

### 2.1. Study Identification and Data Extraction

Retrospective analysis of medical data downloaded from the WoundCareLog database [22] was undertaken. The database recorded data on patients who visited one of 195 cooperative hospitals across China for wound-related diseases, from January 2018 to July 2020. The information on patients’ general features and lower extremity wounds was collected. Specifically, the general features included the patient’s name, gender, age, body height, body weight, home address, hospital department, first admission time, comorbidities, chief complaint, auxiliary examination results, past medical history, and diagnosis. The information regarding diabetic foot ulcers included wound location, depth, duration, wound photographs. The foot ulcer was defined as a full-thickness lesion below the ankle. The medical records in the WoundCareLog database were all uploaded by doctors and nurses in 195 cooperative hospitals across China, following unified standards. All personnel involved in the data collection were trained on several occasions (including site visits) in the use of the case record forms and the techniques required to obtain the data. All case record forms were uploaded to the WoundCareLog database by WoundCareLog APP.

### 2.2. Inclusion and Exclusion Criteria

The diagnostic criteria of diabetes mellitus (DM) were based on the 1999 WHO diagnostic criteria. Wounds were in the lower extremity (below the ankle) and catered for based on the Wagner system of grades 1–5. Patients should have had CT angiography of the injured lower extremity 1 to 3 months prior to the wound evaluation. This study excluded patients with the following characteristics: patients with incomplete data, such as age, weight, height, and so on; and patients with skin lesions caused by other diseases, such as SLE and psoriasis.

### 2.3. Definition of Wound Grading

A diabetic foot ulcer was defined as a skin lesion below the ankle. Diabetic foot ulcers were graded according to the Wagner Score system: grade 0 (skin lesions absent, hyperkeratosis under or over bony prominences); grade 1 (skin and subcutaneous tissue are injured); grade 2 (deeper lesions may penetrate to the bone, tendon, or joint); grade 3 (deep tissues are always involved and osteomyelitis may be developed); grade 4 (gangrene of some portion of the forefoot); and grade 5 (the entire foot is gangrenous) [23]. Patients with DFUs of grade 0 were excluded, because of no skin lesion present. The judgment of wound grading was made by the first-visiting doctors or nurses of cooperative hospitals. This would be checked and modified by wound healing experts after uploading to the WoundCareLog database.

### 2.4. CT Scanning Protocol and Contrast Material Injection Protocol

All patients were scanned with three types of multiple-detector row CT scanners: 16- and 64-multiple-detector CT scanners (Siemens, Erlangen, Germany); 64-General Electric CT scanners (GE Healthcare, Milwaukee, WI). The scanning range for lower extremity CTA was from the top of the liver to the end of the feet, in the craniocaudal direction. The detailed scanning parameters are provided in the Appendix A.

A total volume of 120 mL of nonionic contrast material (Iohexol [Omnipaque 300 mg/mL, Daiichi-Sankyo]; Iodixanol [Visipaque 320 mg/mL, Amersham Health]; or Iopromide [Ultravist 300 mg/mL, Schering]) was injected through a 20-gauge catheter into the antecubital vein with a power injector. The detailed injection protocols are provided in the Appendix A.

### 2.5. CT Angiography Images Assessment

Two experienced radiologists processed the data of patients’ lower-extremity CT angiography to observe the presence or absence of lower-extremity arterial stenosis and the extent of their stenosis. For those with arterial stenosis or occlusion, the affected lower extremity arteries were divided into ten segments: abdominal aorta, common iliac artery, external iliac artery, femoral artery, deep femoral artery, popliteal artery, anterior tibial artery, posterior tibial artery, peroneal artery, and dorsalis pedis artery. The ten segments are sequentially coded as the unordered categorical variable from 1 to 10 and are then imported into the input layer to participate in the construction of the MLP model. According to the degree of lower extremity arterial stenosis, there are four degrees: degree 0 (no stenosis), degree 1 (≤50% stenosis), degree 2 (>50% stenosis but not occluded), and degree 3 (occlusion). In addition, the degree of lower extremity arterial stenosis was coded as an ordinal categorical variable. The presence of arterial calcification was observed from CTA images and coded as a binary variable.

### 2.6. Artificial Neural Network Model

The artificial neural network model was created using the SPSS 26.0 statistical software (SPSS Inc., Chicago, IL, USA). The multilayer perceptron (MLP) algorithm was selected. The MLP is a type of feedforward artificial neural network that has been applied in diverse fields. It can be used to build efficient classifier algorithms for discriminating data that is not linearly separable. The MLP is composed of three layers: the input layer, the hidden layer, and the output layer. To learn the complex relationship between the inputs and outputs, the MLP ANN used predictive factors from the input layer (age, gender, body mass index, duration of diabetes mellitus, duration of a diabetic foot ulcer, limb symptoms, degree of lower-extremity arterial stenosis, segment of lower-extremity arterial stenosis, arterial calcification, and comorbidities) and the output layer (low or high Wagner Score). The output layer consists of two neurons, the learning rate is 0.4, the maximum training time is 15 min, and the model training will be finished when a consecutive step is undertaken with no decrease in error. The activation functions of the hidden layer and the output layer were the Hyperbolic tangent function and SoftMax function, respectively.

Patients in our study cohort were completely randomly allocated, 60% selected as the training sample, 20% as the testing sample and the rest, 20%, as the holdout sample. The training sample and the testing sample were used for the establishment of the ANN models. Once the model was trained, the holdout sample was then used to estimate the performance of the model. A logistic regression model was developed as a control for the ANN model. The flow chart was presented in detail Figure 2.

### 2.7. Statistical Analysis

Data analyses, a receiver operating characteristic (ROC) curve, the area under the curve (AUC) [24], and other measures of performance (accuracy, sensitivity, specificity, positive predictive value, and negative predictive value) [25] were performed using SPSS Version 26 (IBM SPSS; Armonk, NY, USA). The continuous variables are presented as the mean ± SD or medians (interquartile ranges), with analysis by a Student’s *t*-test or a Mann–Whitney U test, as appropriate. We utilized the Shapiro–Wilk test to examine the distribution’s normality. The categorical variables are presented as numbers and percentages and compared using a an χ^2^ test or Fisher’s exact test, as appropriate. The correlation analysis was made using Spearman’s rank correlation coefficient. A *p*-value of less than 0.05 defined statistical significance.

## 3. Results

### 3.1. Patients’ Characteristics, and Comparisons between Patients with a Low and High Wagner Score 

A total of 47,438 case data were analyzed, of which, 20,415 patients with lower-extremity wounds (below the ankle) from 195 hospitals across China were confirmed. According to the exclusion criteria, there were 20,212 case data excluded from the confirmed cases progressively. During the study period, eventually, a total of 203 patients met the inclusion criteria. The flow diagram with detailed information is outlined in Figure 2. 

Patients were of older age (67 ± 11 years), predominantly male (69.5%), and with a long duration of diabetes [10 (4–15) years] and diabetic foot ulcer duration [1 (1–3) months]. The comorbidities were present in 85.7% of the patients, and hypertension accounted for the highest percentage (56.7%). Most patients (63.5%) had no obvious limb symptoms. The degree and segment of lower-extremity arterial stenosis was defined according to the artery with the most severe stenosis or occlusion. Most patients had arterial stenosis in degree 2 (32.5%) and degree 3 (40.4%). In addition, the lower extremity arteries the most susceptible to arterial stenosis were the femoral artery (23.6%), anterior tibial artery (18.2%), abdominal aorta (12.8%) and popliteal artery (12.8%). The detailed demographics of study patients are listed in Table 1.

Define patients with a DFU of Wagner grade 1 to 3 as in the low Wagner Score group (n = 138), and grade 4 and 5 as in the high Wagner Score group (n = 65). As shown in Table 2, there were statistically difference between the two groups in terms of age (*p* = 0.000), body mass index (BMI, *p* = 0.000), DM duration (*p* = 0.000), DFU duration (*p* = 0.017), limb symptoms (*p* = 0.003), the degree of lower extremity arterial stenosis (*p* = 0.000) and the segment of lower-extremity arterial stenosis (*p* = 0.008).

### 3.2. Correlation Analysis

The correlation between clinical predictive factors and Wagner Score in patients with a DFU were listed in Table 3, indicating that both the degree (*ρ* = 0.174, *p* = 0.013) and segment (*ρ* = 0.178, *p* = 0.011) of lower extremity arterial stenosis were positively correlated with their Wagner Score, as well as age (*ρ* =0.331, *p* = 0.000), DM duration (*ρ* = 0.343, *p* = 0.000) and DFU duration (ρ = 0.168, *p* = 0.017). In addition, BMI showed a negative correlation (*ρ* = −0.249, *p* = 0.000).

### 3.3. Model Analysis and Model Evaluation

From the results of the ANN model analysis, we found that the standard feed-forward model with two units in one hidden layer provided the optimal network architecture (Figure 1). The receiver operating characteristic (ROC) curve is presented in Figure 3. As listed in Table 4, the accuracy, sensitivity, specificity, positive predictive value (PPV), and negative predictive value (NPV) of the overall ANN model were 91.6%, 92.3%, 93.5%, 87.0%, 94.2%, respectively. Using the holdout sample to evaluate the model, the accuracy, sensitivity, specificity, PPV and NPV were 88.9%, 90.0%, 88.5%, 75.0% and 95.8%, respectively. As presented in Table 5, the area under the ROC curve (AUC) with a 95% confidence interval of the ANN model was 0.955 (0.924–0.986). The performance of the LR model was inferior to the ANN model.

## 4. Discussion

In this retrospective study, we developed an ANN model to support the accurate prediction of prognosis of DFUs based on clinical data (age, gender, BMI, duration of diabetes mellitus, duration of a diabetic foot ulcer, limb symptoms, arterial calcification, and comorbidities) and lower extremity CTA.

Many studies reported that peripheral arterial disease (PAD) is an independent risk for DFUs and lower extremity amputations [26,27]. Although DFUs are commonly reported and analyzed as one clinical entity, there are significant differences between patients with and without PAD [28]. In contrast, the PAD in patients with a DFU has unique characteristics in terms of its distribution [27]. In this study, it was found that even in the low Wagner Score group, patients with severe arterial stenosis (degree 2 or 3) still accounted for the majority (Table 2). This suggests that even if the ulcer is superficial, we should still pay attention to perfecting the lower extremity arterial examination. 

Clinically, diagnosis of PAD can be made by utilizing CTA, digital subtraction angiography (DSA), the ankle–brachial index (ABI), ultrasound, and magnetic resonance angiography (MRA). ABI is the first-line diagnostic test in patients with PAD, because of its non-invasiveness, low cost, and ease of use. However, ABI can only reflect the overall degree of lower-extremity arterial stenosis but not the segment of stenosis and calcification. CTA is an increasingly attractive imaging modality for assessing the severity of the perfusion deficit of lower extremity arteries [14,15]. Because of it being a noninvasive procedure, with lower expenses and radiation dose exposure [29], and high accuracy (overall sensitivity and specificity rates of around 98%) [14,16,17], CTA is more extensive than other imaging modalities in the assessment of hemodynamically significant stenosis and occlusion. In our study, lower extremity CTA could describe the stenosis in two dimensions: the degree and the segment of arterial stenosis. From our results, there were significant differences between the low and high Wagner Score groups in terms of both the segment and the degree of lower extremity arterial stenosis (Table 2). Meanwhile, the results of the correlation analysis showed that both the segment and the degree of arterial stenosis were positively correlated with Wagner score (Table 3). These two results are in excellent consistency with previous studies on DFUs and PAD, and also indicate that lower extremity CTA, as a method of evaluating PAD, can excellently reflect the relationship between PAD and DFUs from two dimensions: the degree and the segment of arterial stenosis.

ANNs are not novel but are underutilized in wound healing. Compared with traditional technique, the advantage of an ANN lies in the non-linear analysis of complex data in the prediction of results we focus on. As a neural classifier, the MLP algorithm is widely used in diverse fields of medicine [30,31,32,33], and it was utilized in this study to provide a classification for a low or high Wagner Score. To develop the ANN model in this study, 80% of the entire dataset was used as the training sample and the test sample, respectively, in a 3:1 ratio. Since it was not involved in the ANN model construction, the remaining 20% of the dataset was used as the holdout sample for validating the model after it was formed, which can be a more realistic evaluation of the model’s prediction performance. For its overall accuracy of 91.6%, as shown in Table 4, the results assessed using the MLP algorithm in this study can be deemed dependable and accurate in predicting the prognosis of a DFU. From our results (Figure 3, and Table 4 and Table 5), among the model evaluation metrics (ROC, accuracy, sensitivity, specificity, PPV, NPV, and AUC), the generalized linear model performs much inferiorly in this study, compared with the ANN model. Following the principle of Occam’s Razor, we want to solve practical clinical problems with minimum complexity. From the results, there was a significant causality between CTA performance and DFU prognosis, but not a simple linear correlation in terms of values. Currently, there are no studies in the related research field that can predict the occurrence and prognosis of a DFU with high accuracy by simple and easily available predictors, and our study fills this gap.

There is also some limitation to this study. The data for our study was obtained from the WoundCareLog database, which collected cases from 195 cooperative hospitals of different levels across China. The data were recorded and organized by personnel, who came from different hospitals, and then uploaded to the database. Most cases are from patients accepting treatment in outpatient departments. Part of the cooperative hospitals did not equip the specialized equipment to measure the ABI of patients, leading to a massive absence of ABI data. Therefore, the ABIs were not included in the ANN model as a predictive factor. 

In summary, our findings may help clinicians in the early assessment of the wounds and predict the prognosis for patients with diabetic foot ulcers using clinical data and lower extremity CTA, which is easily obtained from clinical practice, and may help patients in reducing the rate of lower extremity amputations.

## 5. Conclusions

The ANN model can reliably predict the prognosis of a DFU based on the patient’s age, gender, BMI, duration of diabetes mellitus, duration of a diabetic foot ulcer, limb symptoms, arterial calcification, comorbidities, and lower extremity CTA. The MLP classifier achieved a 94.1% accuracy in patient classification, which may contribute to reducing the rate of lower extremity amputations of patients with DFUs. This study supports the superior performance of the ANN model used to predict DFU occurrence and prognosis.

## Figures and Tables

**Figure 1 diagnostics-12-01076-f001:**
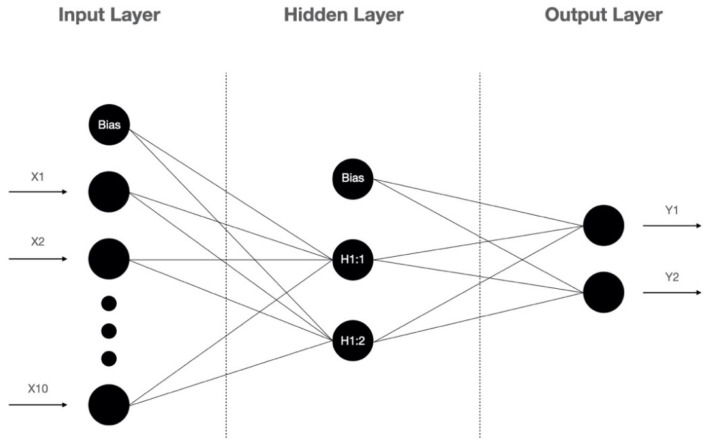
The architecture of the artificial neural network.

**Figure 2 diagnostics-12-01076-f002:**
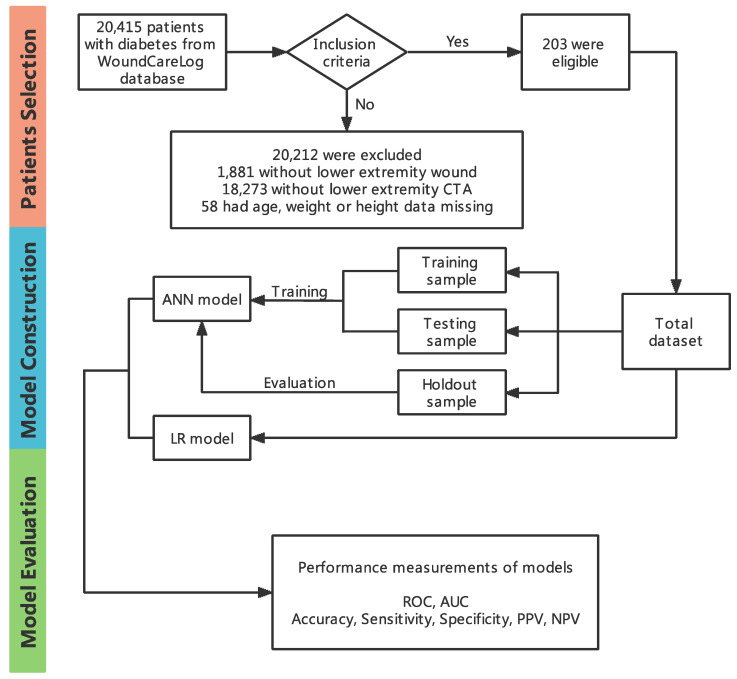
Flow chart of patient selection, model construction and model evaluation.

**Figure 3 diagnostics-12-01076-f003:**
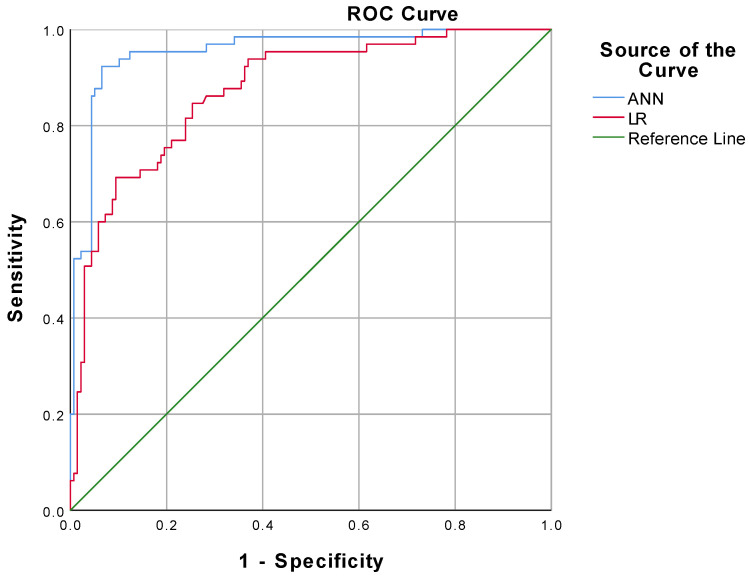
Comparison of ROC curve between ANN model and LR model.

**Table 1 diagnostics-12-01076-t001:** Demographic and clinical characteristics of study patients.

Characteristics	Classification	Total (n = 203)
Gender (n [%])	Male	141 (69.5)
	Female	62 (30.5)
Age (years)		67 ± 11
BMI (kg/m^2^)		23.9 (22.4, 26.4)
DM duration (years)		10 (4, 15)
DFU duration (years)		1 (1, 3)
Limb symptoms (n [%])	Asymptomatic	129 (63.5)
	Mild or moderate claudication	25 (12.3)
	Severe claudication	22 (10.8)
	Critical limb ischemia	27 (13.3)
Degree of lower extremity arterial stenosis	Degree 0	19 (9.4)
	Degree 1	36 (17.7)
	Degree 2	66 (32.5)
	Degree 3	82 (40.4)
Segment of lower extremity arterial stenosis	No stenosis	19 (9.4)
	Abdominal aorta	26 (12.8)
	Common iliac artery	19 (9.4)
	External iliac artery	9 (4.4)
	Deep femoral artery	8 (3.9)
	Femoral artery	48 (23.6)
	Popliteal artery	26 (12.8)
	Anterior tibial artery	37 (18.2)
	Posterior tibial artery	5 (2.5)
	Peroneal artery	3 (1.5)
	Dorsalis pedis artery	3 (1.5)
Arterial calcification	No	67 (33.0)
	Yes	136 (67.0)
Comorbidities	No comorbidity	40 (19.7)
	Cerebral vascular accident	50 (24.6)
	Dyslipidemia	26 (12.8)
	Hypertension	115 (56.7)
	Ischemic heart disease	71 (35.0)
	Nephropathy	22 (10.8)
	Retinopathy	10 (4.9)
	Peripheral neuropathy	58 (28.6)

NOTE: BMI: body mass index; DM duration: duration of diabetes mellitus; DFU duration: duration of diabetic foot ulcer.

**Table 2 diagnostics-12-01076-t002:** Comparison of demographic and clinical characteristics between the low and high Wagner Score groups.

Characteristics	Low Wagner Score	High Wagner Score	*p* Value
Patients (n)	138	65	—
Gender (n [%])			0.304
Male	99	42	
Female	39	23	
Age (years)	64 ± 11	72 ± 10	0.000 **
BMI (kg/m^2^)	24.8 (22.6, 26.9)	23.4 (21.5, 24.7)	0.000 **
DM duration (years)	7 (3, 11)	11 (8, 24)	0.000 **
DFU duration (years)	1 (1, 2)	2 (1, 6)	0.017 *
Limb symptoms (n [%])			0.003 **
Asymptomatic	89	40	
Mild or moderate claudication	18	7	
Severe claudication	8	14	
Critical limb ischemia	23	4	
Degree of lower extremity arterial stenosis			0.000 **
Degree 0	18	1	
Degree 1	32	4	
Degree 2	34	32	
Degree 3	54	28	
Segment of lower extremity arterial stenosis			0.008 **
No stenosis	18	1	
Abdominal aorta	24	2	
Common iliac artery	10	9	
External iliac artery	6	3	
Deep femoral artery	4	4	
Femoral artery	31	17	
Popliteal artery	13	13	
Anterior tibial artery	25	12	
Posterior tibial artery	2	3	
Peroneal artery	3	0	
Dorsalis pedis artery	2	1	
Arterial calcification			0.081
No	51	16	
Yes	87	49	
Comorbidities			0.113
No	23	17	
Yes	115	48	

NOTE: BMI: body mass index; DM duration: duration of diabetes mellitus; DFU duration: duration of diabetic foot ulcer; * *p* < 0.05; ** *p* < 0.01.

**Table 3 diagnostics-12-01076-t003:** Spearman’s rank correlation analysis between Wagner Score and predictors.

Variables	Spearman’s Coefficient (ρ)	*p* Value
Gender	−0.072	0.306
Age	0.331	0.000 **
BMI	−0.249	0.000 **
DM duration	0.343	0.000 **
DFU duration	0.168	0.017 *
Comorbidity	−0.111	0.114
Limb symptoms	0.009	0.903
Degree of lower extremity arterial stenosis	0.174	0.013 *
Segment of lower extremity arterial stenosis	0.178	0.011 *
Arterial calcification	0.122	0.082

NOTE: BMI: body mass index; DM duration: duration of diabetes mellitus; DFU duration: duration of diabetic foot ulcer; * *p* < 0.05; ** *p* < 0.01.

**Table 4 diagnostics-12-01076-t004:** Model evaluation and performance matrices.

Performance Matrix	Formula	ANN (%)	ANN Holdout (%)	LR (%)
Accuracy	$\frac{\mathrm{TP}+\mathrm{TN}}{\mathrm{TP}+\mathrm{TN}+\mathrm{FP}+\mathrm{FN}}$	91.6	88.9	82.8
Sensitivity	$\frac{\mathrm{TP}}{\mathrm{TP}+\mathrm{FN}}$	92.3	90.0	69.2
Specificity	$\frac{\mathrm{TN}}{\mathrm{TN}+\mathrm{FP}}$	93.5	88.5	90.6
PPV	$\frac{\mathrm{TP}}{\mathrm{TP}+\mathrm{FP}}$	87.0	75.0	77.6
NPV	$\frac{\mathrm{TN}}{\mathrm{TN}+\mathrm{FN}}$	94.2	95.8	92.5

NOTE: ANN: artificial neural network; ANN Holdout: Holdout sample evaluate the performance of ANN model; LR: logistic regression; TP: true positives; TN: true negatives; FP: false positives; FN: false negatives; PPV: positive predictive value; NPV: negative predictive value.

**Table 5 diagnostics-12-01076-t005:** The AUC of ANN and LR model.

	AUC	S.E.	95% Confidence Interval	
			Lower Bound	Upper Bound
ANN	0.955	0.016	0.924	0.986
LR	0.874	0.026	0.823	0.925

NOTE: ANN: artificial neural network; LR: logistic regression; S.E.: standard error.

## Data Availability

The data presented in this study are available on request from the corresponding author. The data are not publicly available due to privacy reasons.

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
