# Peer review of "CT-Angiography-Based Outcome Prediction on Diabetic Foot Ulcer Patients: A Statistical Learning Approach"

_diagnostics, 2022, doi:10.3390/diagnostics12051076_

Round 1
Reviewer 1 Report
The topic of the presented manuscript is important, relevant, and original. This is a well written manuscript but some issues could be improved.
-
Introduction seems enough but could achieve a deeper knowledge about state of art.
- The research hypothesis are not described
- Where and how were the subjects recruited? Please, explain it with more detail
-
The complete chronology is missing.
-
Statistical analysis and results are clear enough.
- Discussion section seems confused, needs to be rewritten in some paragraphs
Reviewer 2 Report
Please include CT angiography images to explain how the ten segments are included in 4 degrees.
For this reviewer is not clear how the MLP considers the segment of lower-extremity arterial stenosis?,
the arterial calcification grade is related to the 4 degrees as the authors mentioned, this grade was given by two experienced radiologists, my question is if this value is given by a radiologist or it can be terminated by another technique?. This is because if a radiologist needs to analyze the images he/she can give a prediction
Round 2
Reviewer 2 Report
thank you for your answers